# Spectral signatures of a unique charge density wave in Ta₂NiSe₇

Matthew D. Watson [1] ✉, Alex Louat [1], Cephise Cacho [1], Sungkyun Choi[2,3],
Young Hee Lee [2,4], Michael Neumann[2,3] & Gideok Kim[2,3]

Charge Density Waves (CDW) are commonly associated with the presence of near-Fermi level states which are separated from others, or "nested", by a wavevector of **q**. Here we use Angle-Resolved Photo Emission Spectroscopy (ARPES) on the CDW material Ta₂NiSe₇ and identify a total absence of any plausible nesting of states at the primary CDW wavevector **q**. Nevertheless we observe spectral intensity on replicas of the hole-like valence bands, shifted by a wavevector of **q**, which appears with the CDW transition. In contrast, we find that there is a possible nesting at **2q**, and associate the characters of these bands with the reported atomic modulations at **2q**. Our comprehensive electronic structure perspective shows that the CDW-like transition of Ta₂NiSe₇ is unique, with the primary wavevector **q** being unrelated to any low-energy states, but suggests that the reported modulation at **2q**, which would plausibly connect low-energy states, might be more important for the overall energetics of the problem.

Charge density wave (CDW) instabilities, originally formulated by Peierls for a single 1D band at half-filling, can be naively generalised to real materials by the concept of Fermi surface nesting (FSN)[1–3]. If some portions of the Fermi surface can be mapped to others by a wavevector **q**, a joint electronic and lattice instability with that periodicity can occur when a finite electron-phonon coupling is considered. However it has been argued that the FSN picture is very often insufficient to understand the emergence of charge density waves (CDWs) in real materials, and that the momentum-dependence of the electron-phonon coupling is equally, if not more, important[4–7]. Still, for the CDW to be energetically favourable, one would generically expect at least some low-energy states to be separated by the CDW wavevector **q**, in order for their hybridisation in the modulated phase to lead to an overall electronic energy gain.

At first glance, the structural phase transition in Ta₂NiSe₇[8,9] appears superficially similar to prototypical FSN-driven CDWs as found in e.g., ZrTe₃[10,11], with a moderate transition temperature of ~63 K, incommensurate periodicity[9], a sharp bump-shaped anomaly in resistivity[12,13], and sensitivity of $T_c$ to disorder[12,14,15]. Several literature reports therefore naively assume the FSN picture to be applicable, and

the observed modulations at **q**[9] and **2q**[16] have been sometimes referred to as "$2k_F$" and "$4k_F$", respectively[14]. However, there is actually a lack of support for this scenario from an electronic structure perspective. Early tight-binding models[17] and recent DFT calculations[13,18], as well as transport measurements[12,13], agree that the starting point is a semi-metal with small hole- and electron-like Fermi surfaces, very far from the Peierls paradigm of a single metallic band at half-filling. Furthermore, the Fermi surfaces inferred from semimetallic-like magneto-transport behaviour[12] and calculations[12,13,18] are small, seemingly inconsistent with the ordering at a large wavevector of **q** = (0,0.483,0). Thus, the driving force for the incommensurate CDW in Ta₂NiSe₇ cannot be easily ascribed to a FSN-driven CDW, while the $2q$ structural modulation[16] is another intriguing component, previously only reported in organics[19].

In this work, we reveal the 3D Fermi surface of Ta₂NiSe₇ using high-resolution ARPES and confirm the scenario of small hole- and electron-like Fermi surfaces. Importantly, we identify the total absence of any plausible nesting of states at **q**. We nevertheless observe a backfolding of valence bands in the CDW phase by a wavevector of **q**, with the backfolded bands appearing in a projected bandgap. Thus

---

[1]Diamond Light Source Ltd, Harwell Science and Innovation Campus, Didcot OX11 0DE, UK. [2]Center for Integrated Nanostructure Physics (CINAP), Institute for Basic Science (IBS), Suwon 16419, Republic of Korea. [3]Sungkyunkwan University, Suwon 16419, Republic of Korea. [4]Department of Energy Science, Sung-kyunkwan University, Suwon 16419, Republic of Korea. ✉e-mail: matthew.watson@diamond.ac.uk

$Ta_2NiSe_7$ exhibits the extremely peculiar phenomenology of a prominent backfolding of spectral weight with a wavevector that does not seem to connect any low-energy states. However, we find that there is a possible nesting at **2q**, and associate the characters of these bands with the reported atomic modulations at **2q**. The results lead us to speculate that the CDW may be best understood as a unique instability involving both **q** and **2q**, suggesting an intricate and unique microscopic mechanism that qualitatively differs from the paradigmatic CDW materials.

## Results

The structure of $Ta_2NiSe_7$ consists of three distinct chains, as shown in Fig. 1a. The Ta1 atoms have bicapped trigonal prismatic coordination (BTP), similar to trichalcogenides such as $TaSe_3$[20], forming double-chain units. The Ta2 sites take octahedral (OCT) coordination (as in $1T\text{-}TaSe_2$), and also form double chains of edge-sharing octahedra, akin to those found in $Ta_2NiSe_5$. In between, Ni atoms form chains of extremely distorted octahedra (d-OCT) with substantially varying bond lengths. The seven Se sites are all inequivalent, leading to a rich structure in measurements of the Se 3d core levels (Supplementary Fig. 1).

While possessing the quasi-1D chain structures along $b$, $Ta_2NiSe_7$ can also be considered a 2D material, with a staggered van der Waals gap which yields neutral cleavage in the $b - c$ plane. However, the sawtooth-like structure gives an intrinsically three-dimensionality to the system, and some relatively short interlayer Se–Se distances allow for some interlayer hopping, meaning the 3D character is also important, both structurally and electrically. Thus one might fairly describe the dimensionality as "1+1+1D", or a "dimensional hybrid"[21]. The layers stack according to the $a$ axis, orthogonal to $b$ but not to $c$, and the overall structure is described by a centered monoclinic space group $C2/m$ (number 12).

Substitution of Ni for Pt to form $Ta_2PtSe_7$ was reported in the original synthesis paper of Sunshine and Ibers[8], but no further stoichiometries have been reported in this "217" structure. Structural similarities with $FeNb_3Se_{10}$ were previously mentioned[9,15], but a more contemporary analogy is with $Ta_4Pd_3Te_{16}$, recently discovered to exhibit both an incommensurate CDW and superconductivity[21–24], which has the same space group as $Ta_2NiSe_7$ and similarly contains two distinct Ta chains.

We measured the resistivity of our samples to characterise the $T_c$ and their overall quality. For the sample presented in Fig. 1b, we find $T_c$, defined by the peak in $d\rho/dT$, at 59.5 K, and a residual resistivity ratio (RRR) of 5.2; a second sample (Supplementary Fig. 7) showed slightly lower values of 58.8 K and 4.54 respectively. These values align well with the literature[12] and indicate a good sample quality, albeit that some previous samples achieved slightly improved purity with $T_c$ up to 62.5 K[13] or 63 K[14]. There is a slight hysteresis in the temperature-dependent resistivity, which has been also observed in previous measurements[12,15] and discussed as an impurity-related effect[25].

To give an overview of the electronic structure, we present the calculated density of states in Fig. 1c. The occupied states have primarily Se 4p and some Ni 3d character. Interestingly, the DOS exhibits a pronounced dip near $E_F$, although remaining ungapped. Conceptually, this can be attributed to a slight overlap of the topmost Se/Ni-derived valence band and the bottom-most Ta2-derived conduction bands[17]. For the rest of this paper we focus on these low-energy states at and near $E_F$, but we emphasise that the global picture for the normal state electronic structure of $Ta_2NiSe_7$ is certainly a semimetal.

For an in-depth understanding of the 3D Fermi surface, it is first necessary to consider the somewhat complex geometry of the Brillouin zone, shown in Fig. 2a. From the $\Gamma$ point, a path along the chain direction, $k_{ch}$ direction first intersects the Brillouin zone boundary and

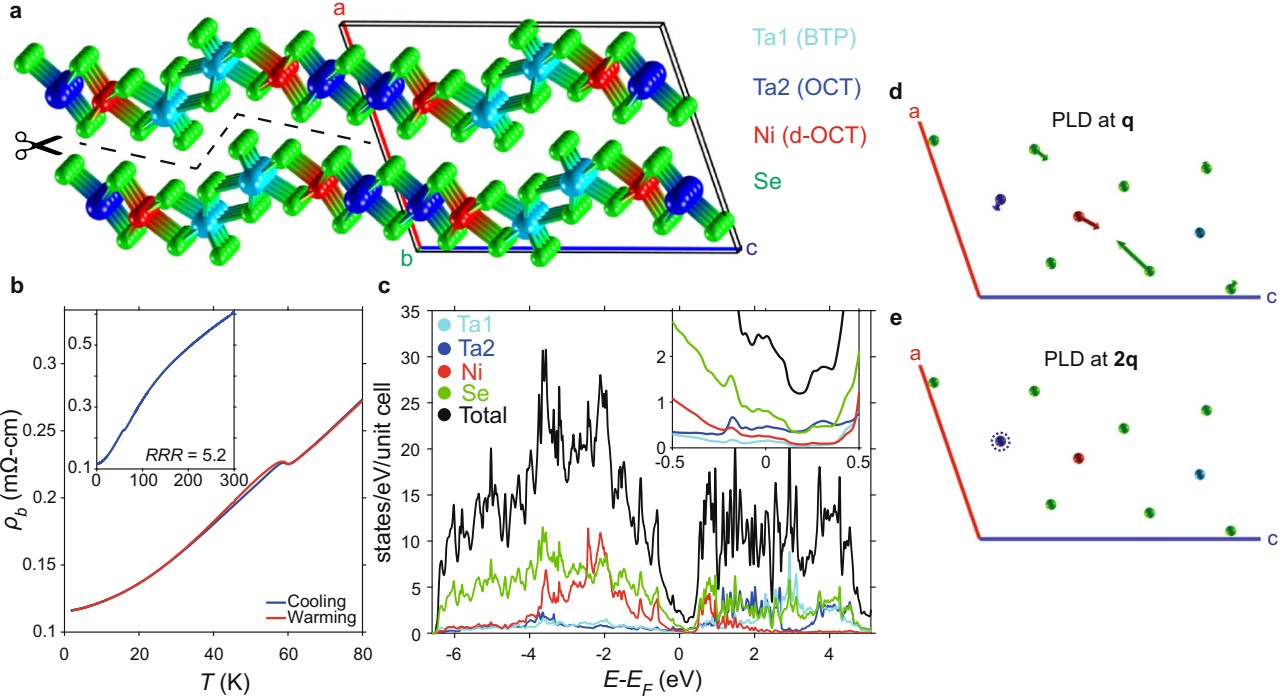

**Fig. 1 | Structural and physical properties of $Ta_2NiSe_7$. a** Crystal structure of $Ta_2NiSe_7$. The dashed line is the cleavage between Se atoms in adjacent layers. The chains direction is $b$, and the conventional unit cell is drawn. The metal coordination geometries are: Ta1: bicapped trigonal prismatic (BTP); Ta2: octahedral (OCT); Ni: distorted octahedral (d-OCT). **b** Resistivity measurement; the anomaly establishes a $T_c$ of 59.5 K in this sample. Inset shows the resistivity curve up to 300 K, from which a residual resistivity ratio (RRR) of 5.2 is extracted. **c** Calculated density of states, showing a pronounced dip around $E_F$ highlighted in the inset, but no gap. **d** Projection of the lattice in the ac plane, showing the periodic lattice distortion at **q**, predominantly transverse displacement of the Ni and Se2 atoms, and **e** at **2q**, involving a longitudinal displacement of the Ta2 atoms (see SM for in-plane projections).

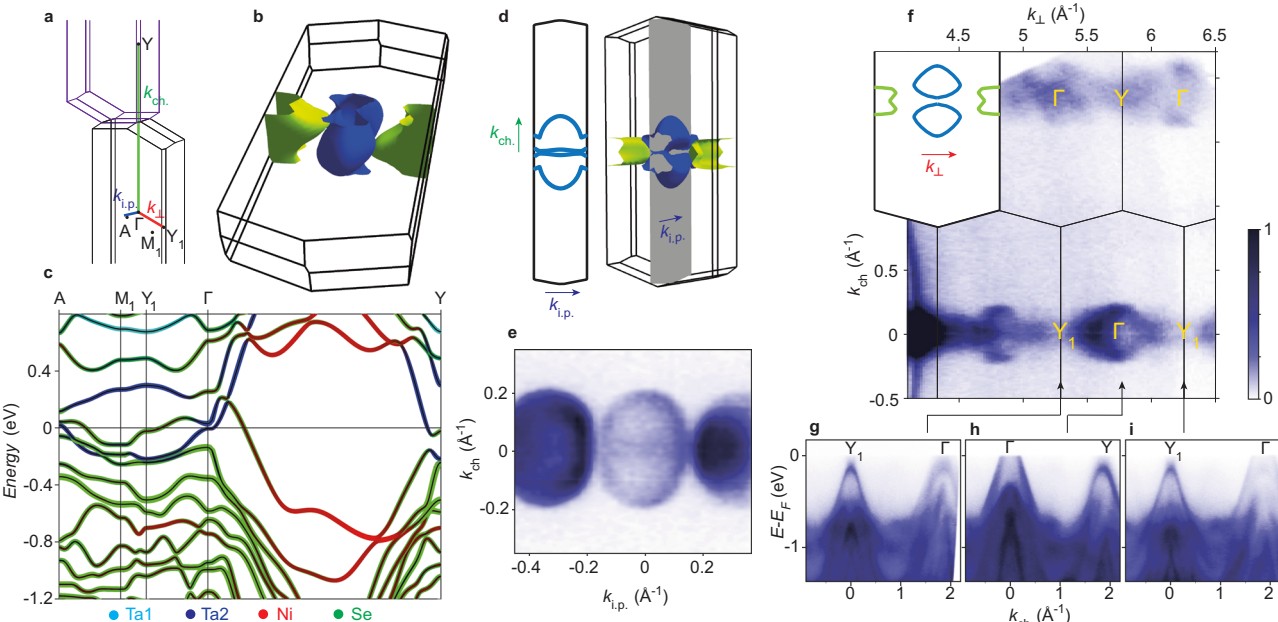

**Fig. 2 | 3D electronic structure of Ta₂NiSe₇. a** Labelled Brillouin zone, showing also the three experimentally accessible $k$ directions: $k_c h$. along the chains, $k_{i.p.}$ accessed by in-plane rotation, and $k_\perp$ accessed by photon energy dependence. The equivalence of the Y and Y1 points is emphasised by plotting an adjacent zone (purple). **b** 3D Fermi surface; the blue Fermi surface is hole-like, while the green is electron-like. **c** Calculated band dispersions with projection of the dominant atomic character. **d** Calculated Fermi surface in the $k_{i.p.} - k_{ch}$ plane, and **e** corresponding experimental measurement, i.e., the Fermi surface map obtained by rotation of the sample. **f** Fermi surface in the $k_\perp - k_{ch}$ plane, obtained at $T = 10$ K from $hv$-dependent data (45–160 eV) using an inner potential of 17 eV. **g–i** Selected high-symmetry dispersions extracted from the $k_\perp$ data.

then encounters the Y point at $(0,2\pi/b,0)$. Note that although we use the notation of the conventional unit cell for describing the real space lattice and the CDW ordering wavevector, for convenience and consistency with previous studies, in $k$-space it is essential to consider the primitive reciprocal lattice vectors and correct Brillouin zone. Importantly, Fig. 2a shows that the Y point, outside of the first Brillouin zone, is related to a formally equivalent point Y₁ on the boundary of the first Brillouin zone, by a primitive reciprocal lattice vector translation.

We align our samples such that the $k_{ch}$ direction is parallel to the entrance slit of our analyser. Then the normal protocol for Fermi surface mapping, i.e., rotating the sample perpendicular to the analyser entrance slit, corresponds to a map in the $k_{i.p.}$-$k_{ch}$ plane. Here $k_{i.p.}$ is the in-plane direction, parallel to the $c$ axis of the real space structure, however in reciprocal space this is not any kind of high-symmetry direction, instead corresponding to the cut shown by the 2D and 3D projections of the calculated Fermi surface in Fig. 2d. With the photon energy tuned to a bulk Γ point, and remaining within the first Brillouin zone, we can use this geometry to nicely map the larger hole-like Fermi surface around Γ in Fig. 2e, revealing a shape consistent with the calculations. Since $k_{i.p.}$ is not a high-symmetry direction in $k$-space, however, this direction is somewhat awkward to make use of, as it will not strictly intersect any other high-symmetry points, and it becomes an even more complicated cut to interpret outside of the first Brillouin Zone. A related feature of the Ta₂NiSe₇ structure is that there is no rotational symmetry about the normal to the the cleavage plane, nor a mirror plane perpendicular to $k_{i.p.}$, i.e., positive and negative $k_{i.p}$ are not equivalent. Thus, azimuthal rotation of the sample by 180°−typically an innocuous operation on needle-like samples−does not result in equivalent spectra, as explained further in Supplementary Fig. 2.

Since the calculations shown in Fig. 2b, c predict a 3D electronic structure, and additionally $k_{i.p.}$ is not the most intuitive direction to probe in this case, it is especially important to make full use of the out-of-plane direction, $k_\perp$, which at normal emission corresponds to a path along Γ-Y₁-Γ. Relying on the nearly free electron final state model, (which seems to work adequately in this case), $k_\perp$ becomes

experimentally accessible by varying the photon energy in our ARPES experiment. After performing the standard conversion to $k_\perp$, in Fig. 2f we present the Fermi surface as projected in the $k_\perp - k_{ch}$ plane. We find periodic structure, matching the expected stacking of the 3D Brillouin zones for this cut. The data reveals a larger Fermi pocket centered around each Γ point. Taken together with Fig. 2e, this confirms the 3D character of the hole pocket by its dispersion in the $k_{ch}, k_{i.p.}$ and $k_\perp$ directions.

The data in Fig. 2f also shows streaks of intensity near the Y and Y₁ points, consistent with the electron-like Fermi surface expected from calculations. The corresponding dispersions are shown in Fig. 2g–i, where at the Y point, and more clearly at the Y₁ points, we observe a fully occupied hole-like band, above which a small electron-like dispersion reaches up to $E_F$. Despite best efforts, this band always appears somewhat broadened and poorly resolved compared with the sharp hole bands at Γ. We understand this principally as an effect of $k_\perp$-averaging[26,27], not helped by a weak matrix element, but an electron-like band is consistently observed around each Y or Y₁ point.

Overall we find that the experimental closely matches the calculated 3D Fermi surface, and qualitatively at least, we confirm the scenario that Ta₂NiSe₇ is a semimetal with relatively small, 3D, hole- and electron-like Fermi pockets, as implied by transport measurements[12,13]. There are a few notable differences, however, between the calculations and the experiment. Although the band structure at the Γ point seems to agree well with the calculation, the outermost $k_F$ of the hole pocket $k_{F,ch}$ is smaller in experiment (0.18 Å⁻¹) than in the calculation (0.27 Å⁻¹). Moreover, the agreement at the Y (or Y₁) point is less good; the calculation implies that the Ta2-derived conduction band dips below the uppermost valence band at Y, opening a hybridisation gap, whereas experimentally it appears as if the conduction band sits entirely above the topmost valence band (see Supplementary Fig. 3). These differences might be explained, at least partially, by the fact that DFT typically underestimates the band gaps in semiconductors, and by extension, overestimates the band overlap in semimetals. Here we chose the mBJ potential, developed specifically for better estimates of

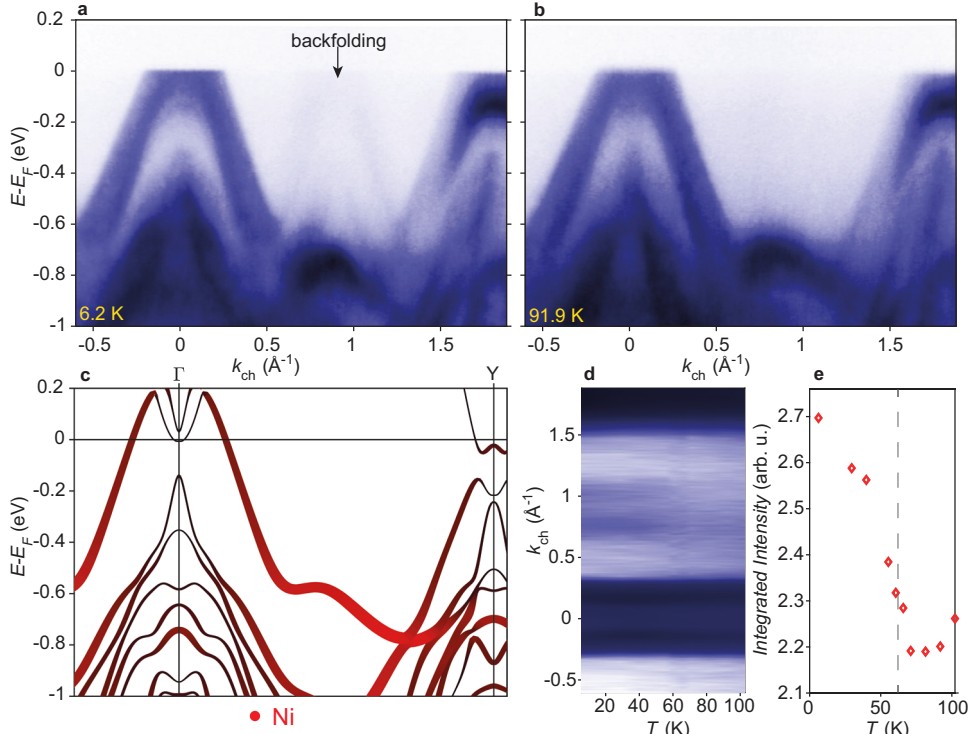

**Fig. 3 | Evidence for backfolded spectral weight below $T_c$. a, b** Dispersion along $k_{ch}$ taken at $hv = 79$ eV, at temperatures above and below $T_c$, highlighting the backfolded spectral weight at low temperatures. **c** DFT calculation along the Γ-Y path, highlighting the Ni character that is dominant on the band which is also observed to be backfolded. **d** Temperature-dependent momentum distribution curve (MDC) integrating between −0.15 and −0.11 eV. **e** Integrated intensity between $0.71 < k < 1.10$ Å$^{-1}$, showing an onset of the backfolded spectral weight close to $T_c$ (dashed line).

the band gaps, which is known to improve the agreement with experimental data in several semimetals and specifically Ta$_2$NiSe$_5$[28], but residual inaccuracies on the -100 meV scale may remain.

We now turn to the principal signature of the CDW in our ARPES measurements, namely the appearance of backfolded spectral weight at low temperatures, shown in Fig. 3a. The backfolded bands appear as near-copies of the bands observed at $\bar{\Gamma}$, with the displacement being consistent with the reported CDW wavevector **q** = (0, 0.483$b^*$, 0). The backfolded bands appear in a projected bandgap between $\bar{\Gamma}$ and $\bar{Y}$, i.e., an empty region where there are no states at any $k_\perp$ above $T_c$, as shown in Fig. 3b. Figure 3d, e shows that the backfolded spectral weight onsets at (or very close to) $T_c$.

In broad terms, the appearance of backfolded bands in ARPES can be either associated with electronic hybridisation, or structural replicas due to an imposed superstructure, in which case either Umklapp scattering of the initial state due to the new periodicity or diffraction of the photoelectron final state can yield similar spectra. A famous example of backfolding due to electronic hybridisation would be TiSe$_2$, where in the 2 × 2 × 2 ground state the hole bands from Γ appear as backfolded at the L point of the Brillouin zone[29]. However in that case the backfolded bands bear the signatures of electronic hybridisation with a flattening of the dispersion around the band maxima, and the intensity of the backfolded bands is bright, but decays away from the band maximum. The replica bands observed in graphene grown on various substrates, however, are usually interpreted in terms of diffraction replicas[30], and typically show approximately uniform intensity along the length replica dispersion, over -eV energy scales.

The observed replica bands here appear over a wide energy scale: they can be traced from the Fermi level down to at least −0.6 eV below $E_F$, where their spectral weight then merges with deeper-lying bands. This high energy scale seems at odds with the temperature scale of the phase transition at 60 K, and furthermore the quantitative ratio of spectral weight which is backfolded is low (-6%). This evidence

suggests that the origin of the backfolded spectral weight is either Umklapp scattering of the initial state, or diffraction of the photoelectron final state. These scenarios are notoriously hard to distinguish[30], and we are unable to make a conclusive statement on the origin in this case.

Independent of the mechanism giving rise to the backfolded spectral weight, in the context of a CDW material, it is unusual and interesting that the bands which are backfolded by the wavevector **q** do not appear to be linked to an electronic hybridisation mechanism. However, it is worth observing that the main band which is backfolded by the wavevector **q**, i.e., the outer hole band at Γ, has dominantly Ni character (Fig. 3c). Interestingly, the calculation in Fig. 3c suggests that the proportion of Ni weight along the band increases at higher binding energies, while the data in Fig. 3a shows a somewhat stronger backfolding at higher binding energies, suggesting a correlation between the Ni character and the degree of backfolding. Indeed, it makes sense for this band to experience Umklapp scattering by **q**, because the band derives from Ni atoms (and some Se) that also undergo a periodic lattice distortion with wavevector **q**.

The fact that the bands that are replicated from Γ appear in a projected bandgap that is completely void of states above $T_c$ underlines the central puzzle of Ta$_2$NiSe$_7$—that there are simply no low-energy states in the high-temperature phase that are linked by the wavevector **q**. We summarise the situation in Fig. 4. Relying on the data presented thus far for the occupied states, and the earlier calculations in Fig. 2c for the unoccupied states, we can state with confidence that there are no states within at least 0.4 eV of the Fermi level on either side that can possibly by linked by the wavevector **q**. Nevertheless, the modulation at **q** observed in previous scattering experiments is manifested here by the backfolding of bands by **q**. However, the experimental electronic structure does provide one more clue that is likely to be linked to the overall understanding of the phase transition. Namely, there are states which can be linked by the wavevector **2q**, and in Fig. 4

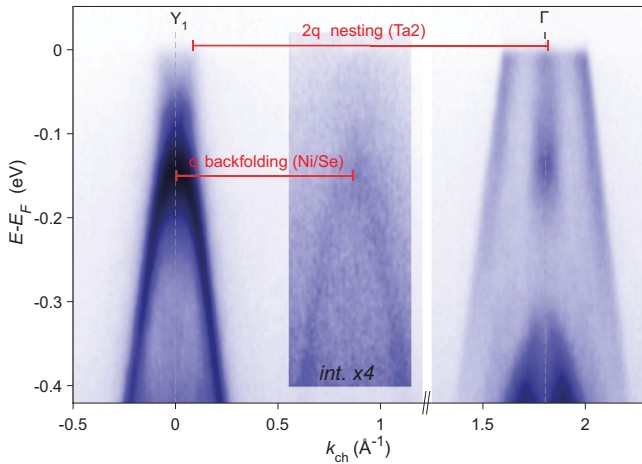

**Fig. 4 | Composite high-symmetry dispersion.** Data taken around $k_{ch} = 0$ at $h\nu =$ 95 eV ($Y_1$ point), and data from a bulk $\Gamma$ point (77 eV) shown with $k_{ch}$ values shifted by 1.8040 Å $= 2\pi/b$, so that the $k$-path of the figure overall is equivalent to $Y_1 - \Gamma$ (same as Fig. 2g, i). A region of enhanced colour scale emphasises the weak intensity of a replica dispersion, corresponding to the fully occupied valence band from Y displaced by a wavevector of $|q| = 0.483(2\pi/b)$. A possible nesting between states with Ta2 character separated by $2q$ is indicated.

we draw a possible connection between two Ta2-derived states: the previously-identified conduction band at $Y/Y_1$, and a peak in the spectral weight at $\Gamma$ that corresponds to another barely-occupied electron-like dispersion (see Fig. 2c and Supplementary Fig. 3). Notably, the only significant atomic modulation at **2q** identified in ref. 16 was the longitudinal motion of the Ta2 sites. Therefore, although we have not explicitly found energy gaps appearing on these states, we are led to speculate that the energy gain that drives the CDW might originate primarily from the **2q** modulation on the Ta chains, rather than the **q** modulation on the Se and Ni sites. Supporting evidence that the **2q** order might play a leading role comes from the observation in ref. 16 that the fluctuations associated with the **2q** order persist up to 200 K, while the fluctuations at **q** can only be seen just above $T_c$. Although a full microscopic understanding is not within the scope of this experimental paper, we suggest that the **2q** ordering, and the Ta2 electron-like dispersions, may be the key to understanding the energetics of this unusual CDW.

## Discussion

Comparing $Ta_2NiSe_7$ with the famous "215" stoichiometry, $Ta_2NiSe_5$, there are similarities in that both are semimetals in their high-temperature phases (at least, according to DFT), and in both cases the conduction band(s) derives from a slight occupation of Ta $5d$ states. However, the nature of the resulting structural instabilities are very different in the two cases. In $Ta_2NiSe_5$ there is a $\mathbf{q} = 0$ orthorhombic-monoclinic structural distortion at a much higher temperature of 326 K in which a hybridisation between hole and conduction states in the low-temperature phase opens an energy gap of ~0.16 eV, with ongoing debate over the role of excitonic interactions in driving the transition[28,31–33]. In contrast, in $Ta_2NiSe_7$ the ordering wavevector **q** (and **2q**) is finite and incommensurate, with the system remaining metallic below $T_c$.

Our measurements were unable to detect direct evidence for any energy gaps opening at the Fermi level. We make the caveat that, although there is no gap resolved at the outer $k_F$ of the hole pocket in the high-symmetry plane where the data is relatively sharp, the 3D character of the hole-like Fermi surfaces might further obscure more subtle signatures especially if located at non-zero $k_\perp$. Furthermore, we cannot exclude a gap forming on the Ta2-derived conduction band at Y, due to the low spectral weight, and subtle energy gaps on the scale

of a few meV cannot be excluded due to the finite energy resolution. Another possibility is that the energy gap might lie primarily above $E_F$[34].

The transport measurements of $Ta_2NiSe_7$ might help to understand the absence of an observable CDW gap. The absolute deviation of the resistivity at $T_c$ is fairly modest, as shown in Fig. 1c, and the carrier densities remain relatively constant through $T_c$[12]. This would seem to imply that the majority of states at $E_F$ remain ungapped and may be relatively unperturbed by the CDW. This situation is not without precedent; $1T\text{-}VSe_2$ has a CDW which also manifests as a bump-like anomaly in resistivity at the onset of the CDW[35], but in that case as well, ARPES results on the bulk material have so far failed to reveal any energy gaps at $E_F$[6].

As for the prediction that $Ta_2NiSe_7$ could be a $Z_4 = 3$ 3D topological insulator[36], we comment that we have not observed any surface states in $Ta_2NiSe_7$. An important consideration is that the DFT-based predictions of ref. 36, using the standard generalized gradient approximation (GGA), substantially overestimate the degree of overlap between the Ni/Se hole-like bands and the Ta2 electron-like bands, compared with either our calculations using the mBJ potential, or with the experimental data, which could potentially alter the topological assignments. Nevertheless, the topological viewpoint on $Ta_2NiSe_7$ remains of interest and further calculations would be desirable, in order to confirm any topological characteristics of the band structure, in combination with the data shown here.

The unusual CDW in $Ta_2NiSe_7$ merits much further examination. For example, besides one study of Ni doping[14], it remains an open question to what extent the CDW can be tuned with chemical substitution, pressure, or strain. Given that the end compound is already reported to be stable[8], studying $Ta_2(Ni_{1-x}Pt_x)Se_7$ seems like a promising new avenue, while the chain-like structures make the system amenable to strain studies, which may be able to tune the band overlap. The early low-temperature scanning tunneling microscopy measurements yielded controversial results[37], but with modern atomic resolution systems one ought to be able to resolve the separate **q** and **2q** modulations residing on the Ni and Ta2 chains. Our interpretation leans heavily on the crystallographic work of ref. 16 which first shed light on the **2q** modulation; reproducing and extending this with a systematic temperature-dependence seems a high priority for future studies. Inelastic X-ray scattering would also be highly informative, in order to establish which (if any) phonon modes soften at $T_c$, and at which **q**. Most of all, our work clearly calls for in-depth ab initio studies to help develop a more microscopic framework for understanding the experimental results, particularly calculations of the electronic susceptibilities and $k$-dependent electron-phonon coupling[4,6,38].

In conclusion, we have experimentally determined the 3D Fermi surface and electronic structure of $Ta_2NiSe_7$. We find small hole- and electron-like pockets, closely resembling our calculations. Importantly, our results show there is no possible nesting of any bands near $E_F$ for the primary CDW wavevector, **q**. Nevertheless, below $T_c$, we observe backfolding of bands with mixed Ni and Se character by the wavevector q, with the spectral weight transfer likely to be of structural origin (i.e., Umklapp scattering of the initial state, or perhaps scattering of the photoelectron final state) rather than reflecting an electronic hybridisation. In contrast, there is a possible nesting of states with Ta2 character with the wavevector **2q**, aligning with the literature result that displacements associated with the the **2q** modulation are dominantly on the Ta2. In the absence of any plausible energy gain from states linked by **q**, we speculate that the **2q** component may play a key role in stabilising the order, although further experimental and theoretical work is called for to fully establish the role of the 2q ordering. However, at the Fermi level, we do not resolve any changes through $T_c$ and the CDW gap remains elusive. Our results establish the CDW in $Ta_2NiSe_7$ as a highly unusual case where the ordering at **q** seems unrelated to any low-energy electronic states, raising the intriguing possibility of a critical role for the **2q** ordering.

## Methods

Single crystals were grown by a chemical vapor transport method with 5 at. % excess selenium as the transport agent. A temperature gradient was introduced to the sealed tube with the cold zone temperature 690 °C and the hot zone temperature 790 °C, and the tube was kept in the furnace for 2 weeks. ARPES measurements were performed on the HR branch of the I05 beamline of Diamond Light Source[39], using photon energies in the range 30–240 eV, at sample temperatures generally below 10 K except for temperature-dependent runs. DFT calculations were performed on the experimental crystal structure[16] within the Wien2k package[40], using the modified Becke–Johnson (TB-mBJ) functional[41] and accounting for spin–orbit coupling. The TB-mBJ potential was used as it generally gives a better estimate of the band overlap in semimetals compared with standard GGA or LDA functionals[41], and specifically gives a reasonable description of $Ta_2NiSe_5$[28]. 3D Fermi surfaces were plotted using FermiSurfer[42].

## Data availability

The data that support the findings of this study are available from the corresponding author upon reasonable request.

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

## Acknowledgements

We thank T.K. Kim and P.D.C. King for insightful discussions. We acknowledge Diamond Light Source for time on beamline I05 under proposal NT31067. We would like to thank Gavin Stenning for help on the PPMS2 instrument in the Materials Characterisation Laboratory at the ISIS Neutron and Muon Source. G.K., M.N. and Y.H.L. acknowledge support from the Institute for Basic Science (IBS-R011-D1). S. C. is supported by the Institute for Basic Science (IBS-R011-Y3) in Republic of Korea.

## Author contributions

Crystals were grown and characterised by G.K., S.C., Y.H.L., and M.N. M.W., A.L., and C.C. performed the ARPES experiments. A.L. performed the resistivity measurements. M.W. analysed the data and wrote the paper, with input from all co-authors.

## Competing interests

The authors declare no competing interests.
