## [Peer Review File · Nature Communications]

REVIEWER COMMENTS

Reviewer #1 (Remarks to the Author):

The manuscript of Watson et al. reports on an electronic study of Ta₂NiSe₇, in which they observe the feature of a CDW at a wavevector q that is not relevant to Fermi surface nesting. I believe that the authors have performed the work properly and the data shown are correct. However, I do not think the authors have shown enough evidence to support their claim and thus not conclusive to be published in Nat. Comm. To be more specific, I do not understand why the CDW wavevector is q while the nesting can be $2q$. The authors have not given a reasonable explanation and thus the manuscript is not convincing.

I would say that the final state effect can be a perfect explanation for the observed phenomenon. If it is the initial state as the authors claim, they need to explain in detail why the final state effect can be excluded in more detail. I could not find any signature of a “hybridization” as the authors have stated.

I would also like to see data from other experimental techniques such as XRD or STM to support that the CDW wavevector is q in the sample they have studied.

Reviewer #2 (Remarks to the Author):

In this work, the authors investigate the electronic structure of the incommensurate charge density wave (CDW) state in the transition-metal chalcogenide Ta₂NiSe₇. Combining the ARPES and DFT calculations, the authors examine the correspondence of the Fermi surface between the experiment and simulation. Then, the authors show the backfolded spectrum emerging at the vector q below the transition temperature T_c . Because this backfolding vector q is not consistent with the nesting vector $2q$ expected by the Fermi surfaces, the authors conclude that Ta₂NiSe₇ has the unconventional nature of the CDW.

This manuscript reports a very peculiar CDW state; its nature is interesting and may attract attention. The ARPES spectrum is clearly presented with the guide of the DFT bands, which is helpful. While there is inaccuracy in the DFT calculation around the Fermi level, the DFT bands show agreements with the ARPES spectrum in the broad range. I think this is an acceptable level of inaccuracy in the first report on the comprehensive electronic structure of this material. I believe this manuscript is promising for publication in Nature Communications. However, there are several

confusing points in the present manuscript, and I would like to suggest possible revisions. My suggestions are listed below.

1) In the present manuscript, it is very hard to understand the real space picture of the CDW. Specifically, the relation between the q lattice displacement and the $2q$ modulation related to Ta2 is confusing. I strongly require the authors to visualize the real space picture (or lattice structure) in the low-temperature CDW phase even though it is schematically indicated in the supplemental materials.

2) I expect that the authors can obtain some insights into the backfolded bands in the CDW phase using DFT calculations if the crystal structure of the low-temperature phase has been reported (e.g., in Ref. [16]). I understand that the simulation of the incommensurate CDW is very difficult. However, because the modulation vector $q = 0.483b^*$ is close to $q = 0.5b^*$, a DFT calculation assuming a virtual short-range ($q = 0.5b^*$) superlattice structure may give us meaningful insights. I think this can be a nice approximation because incommensurate or not is not so important for the emergence of the backfolded band at q . If this kind of approximated calculation is possible, I encourage the authors to perform additional calculations.

3) As written in the first paragraph, Fermi surface nesting (FSN) is not always the dominant driving force of the CDW. In Ta₂NiSe₇, lattice instability is probably the main source of the modulation with q . While the evaluation of the lattice dynamics is not easy in this complex material, an examination of the importance of FSN is possible by calculating the bare electron susceptibility from the DFT bands. If the susceptibility is presented (even in the supplemental information), the authors can discuss the contributions from the electronic bands more plausibly.

4) At first glance, I cannot find the meanings of “BTP”, “OCT”, and “d-OCT” in Fig.1(a). They probably mean “bicapped trigonal prismatic”, “octahedra”, and “distorted octahedra”, respectively, as written in the main text. The authors should put their abbreviations behind them [e.g., octahedra (OCT) in the main text].

5) The authors should present the value of the temperature used in the ARPES measurements in the figure caption of Fig.2.

6) On the first line on page 9, the authors mention that both Ta₂NiSe₇ and Ta₂NiSe₅ are semimetals in their high-temperature phases. This is true in the DFT calculations. However, the temperature dependences of the resistivities of Ta₂NiSe₅ and Ta₂NiSe₇ are obviously different, and the gap opening of Ta₂NiSe₅ above T_c is probably under debate [e.g., Nat. Phys. 17, 1024 (2021)]. The

author should express this sentence considerably (for example, “both are semimetals”  “both DFT band structures are semimetallic”).

Reviewer #3 (Remarks to the Author):

Watson et al. present an experimental study of the electronic structure of Ta₂NiSe₇ using high-resolution ARPES which is supported by DFT band calculations. Ta₂NiSe₇ is a less-well-studied CDW material with $T_c \approx 60$ K. Its cousin, the “215” compound (Ta₂NiSe₅) however, has been widely studied because it is an excitonic insulator candidate. Here, the authors investigate the CDW in Ta₂NiSe₇ by mapping the Fermi surface and conclude that no states can be directly “nested” onto one another by the expected q -vector, based on basic geometry arguments. Instead, they claim that some portions of FS can be nested by $2q$, although they do not observe associated bandgaps opening in the reconstructed phase - a key signature of CDW formation. Curiously, they do however observe a backfolded band with reasonable intensity at q . The overall picture is unusual and slightly confusing, and the authors leave many open questions. Previous ARPES works exist e.g. [Ref 18], although the study by Watson et al. is likely to be the most detailed to date.

Overall, I find the manuscript to be very well-written and clearly presented. The main results however are somewhat speculative, relying on clues in previous works and then interpreting the main features of the electronic structure using geometric arguments related to the CDW i.e. by essentially drawing an arrow (like we see in FIG.4) to test if there are states which are separated by a vector of expected length, q . At a basic level, this is certainly a prerequisite for CDW formation, and we see this type of geometric interpretation in the literature often without deeper analysis. Despite this, the authors do present a detailed and engaging discussion, and have clearly considered several scenarios. Since the case of Ta₂NiSe₇ seems to be rather complicated (q vs. $2q$), and some effects are hidden to the current data (e.g., gap opening), further calculations of the k -dependent electron-phonon coupling [Phys. Rev. B 77, 165135] or electronic susceptibility [Phys. Rev. B 88, 035108 (2013)] would be informative. In this way, one could analyse potential contributions at both q and $2q$ and learn more about the dominant mechanism. Related to the hidden gap problem, the authors already touch on the complications of the 3D FS (k -perp.), have they also considered the potential for above- E_f gaps [e.g. Nat. Comms. 12, 6837, (2021)] or is this unlikely in the present situation?

A curiosity: In FIG 3b, at 91.9 K, slightly above T_c , it seems as if there is some weak spectral weight in the region where the backfolded band later appears at 6.2 K. In TiSe₂ (a famous example of intense backfolded CDW bands), spectral weight above T_c was attributed to fluctuations [New J. Phys. 14 075026 (2012)]. On page 8, the authors hint at fluctuations related to q in Ref 16 – are there clues in the authors’ data?

Final minor comment about the use of colloquial or informal language which are not in-keeping with rest of the paper, listed below:

- Page 1: It is perhaps not appropriate to refer to Peierls' 1955 theory as a "toy model" (despite its obvious simplicity).
- Page 5, para 3: "fuzzy": Poorly-resolved? Broadened? Noisy? Poor statistics? These are not only alternative word suggestions, but also have different scientific origins. Perhaps the authors can select the most appropriate option for their supposed k-perp averaging problem.
- Page 6, para 1: "bad underestimation" is subjective. Perhaps "DFT typically underestimates band gaps in semiconductors"

In summary, the work by Watson et al. represents a comprehensive study of the electronic structure of Ta₂NiSe₇ from both experiments and calculations, with excellent data quality (low temperature, high-resolution, photon energy dependent ARPES). The observation of the backfolded back at q is interesting, and similarly the lack of potential 'nesting' at $2q$ and related bandgap expected for a CDW. Although the main conclusions of the paper are not ground-breaking for understanding the CDW in Ta₂NiSe₇ at this stage, the work is still likely to be important to the community and will motivate further study to answer the questions raised. Based on this, I would be inclined to recommend the work for publication, given that the authors carefully consider my comments.

Reviewer #1 (Remarks to the Author):

The manuscript of Watson et al. reports on an electronic study of Ta₂NiSe₇, in which they observe the feature of a CDW at a wavevector q that is not relevant to Fermi surface nesting. I believe that the authors have performed the work properly and the data shown are correct.

However, I do not think the authors have not shown enough evidence to support their claim and thus not conclusive to be published in Nat. Comm. To be more specific, I do not understand why the CDW wavevector is q while the nesting can be $2q$. The authors have not given a reasonable explanation and thus the manuscript is not convincing.

While admitting that we do not yet have a full understanding of the instability, which appears to be a “very peculiar CDW state” (Ref 2), we believe that our manuscript makes some robust, new and significant statements:

- That the electronic structure is 3D, broadly agreeing with DFT (mBJ) calculations, and displays periodicity consistent with the correct Brillouin zone
- That the primary periodic lattice distortion (PLD) at q manifests in photoemission as a backfolded spectral weight below T_c
- That there is no plausible connection between low energy electronic states separated by q
- This makes the CDW quite unusual, as the typical energy gain from gapping states close to EF is simply not available for this q vector.
- That there is a possible connection between states separated by a wavevector of $2q$.

These points are all new, factual in nature, and taken together establish Ta₂NiSe₇ as an unusual CDW that merits much further examination. On top of these, we make some slightly more speculative statements:

- That the states at EF connected (nested) by $2q$ both have Ta₂ character
- That the $2q$ component of the CDW – which involves almost exclusively the Ta₂ atoms - might be linked to the hybridisation of these states
- That the $2q$ CDW might have a key role in understanding the overall energetics of the CDW

While we accept that these ideas might turn out to be over-simplistic, it is currently very early in the discussion of the CDW in this material, and our intention is to provide high quality data that give a backdrop for future discussion, rather than settling the whole mechanism. We are more than happy to engage further with the referee to understand which particular claims of the manuscript that they are not convinced by.

I would say that the final state effect can be a perfect explanation for the observed phenomenon. If it is the initial state as the authors claim, they need to explain in detail why the final state effect can be excluded in more detail. I could not find any signature of a “hybridization” as the authors have stated.

There are two separate issues here. We agree with the referee that “the final state effect can be a perfect explanation” for the observation of backfolded spectral weight. We also agree that the “signature of hybridisation” was not a well-founded claim, and so we have cut this claim and now have a more balanced discussion of origin of the backfolding.

However irrespective of the origin of the backfolded spectral weight, there remains the central claim and conundrum of this paper – namely that the wavevector q does not seem to connect any low-energy states, and therefore the observation of ordering at this wavevector is rather peculiar.

I would also like to see data from other experimental techniques such as XRD or STM to support that the CDW wavevector is q in the sample they have studied.

We agree with the referee in seeing value in future XRD and STM experiments – and indeed we already highlight these techniques as potentially fruitful avenues of investigation in the discussion section. It is fair to say that we rely quite heavily on literature determinations of the q and $2q$ modulations, particularly Ludecke 2000, rather than our own. However, we have performed XRD at room temperature and now include the refinement results in the supplemental material, finding structural parameters consistent with the literature. For this study, three separate samples were measured by resistivity, all showing anomalies at T_s values consistent with the literature. Moreover some of us are involved in a Raman scattering study on this material which shows changes through T_s consistent with an

incommensurate ordering. Most pertinently, we observe the wavevector \mathbf{q} directly in the ARPES data by the observation of backfolded spectral weight. In summary we don't accept that there is a reason to doubt that our samples are comparable with the literature and there is no doubt that there is a CDW at $\mathbf{q}=(0,0.483,0)$.

Reviewer #2 (Remarks to the Author):

In this work, the authors investigate the electronic structure of the incommensurate charge density wave (CDW) state in the transition-metal chalcogenide Ta₂NiSe₇. Combining the ARPES and DFT calculations, the authors examine the correspondence of the Fermi surface between the experiment and simulation. Then, the authors show the backfolded spectrum emerging at the vector \mathbf{q} below the transition temperature T_c . Because this backfolding vector \mathbf{q} is not consistent with the nesting vector $2\mathbf{q}$ expected by the Fermi surfaces, the authors conclude that Ta₂NiSe₇ has the unconventional nature of the CDW.

This manuscript reports a very peculiar CDW state; its nature is interesting and may attract attention. The ARPES spectrum is clearly presented with the guide of the DFT bands, which is helpful. While there is inaccuracy in the DFT calculation around the Fermi level, the DFT bands show agreements with the ARPES spectrum in the broad range. I think this is an acceptable level of inaccuracy in the first report on the comprehensive electronic structure of this material. I believe this manuscript is promising for publication in Nature Communications. However, there are several confusing points in the present manuscript, and I would like to suggest possible revisions. My suggestions are listed below.

We thank the referee for their positive remarks.

1) In the present manuscript, it is very hard to understand the real space picture of the CDW. Specifically, the relation between the \mathbf{q} lattice displacement and the $2\mathbf{q}$ modulation related to Ta₂ is confusing. I strongly require the authors to visualize the real space picture (or lattice structure) in the low-temperature CDW phase even though it is schematically indicated in the supplemental materials.

While appreciating the referee's point, the visualisation is somewhat difficult as there are multiple components and a 3D representation is required, and many unit cells are required to fully present the incommensurate order if the relative phases of the distortion are to be properly accounted for. We now include the projection of the PLD in the ac plane in Fig 1, according to the literature reference of Ludecke 2000, and the corresponding projections in the bc plane in a new figure SM5 in the SM. For convenience we also include the new figure here:

Fig R1 – also added to S.M. – visualising the CDW displacements.

2) I expect that the authors can obtain some insights into the backfolded bands in the CDW phase using DFT calculations if the crystal structure of the low-temperature phase has been reported (e.g., in Ref. [16]). I understand that the simulation of the incommensurate CDW is very difficult. However, because the modulation vector $q = 0.483b^*$ is close to $q = 0.5b^*$, a DFT calculation assuming a virtual short-range ($q = 0.5b^*$) superlattice structure may give us meaningful insights. I think this can be a nice approximation because incommensurate or not is not so important for the emergence of the backfolded band at q . If this kind of approximated calculation is possible, I encourage the authors to perform additional calculations.

We thank the referee for this suggestion and we have added a new section in the supplementary material where we have implemented a $1 \times 2 \times 1$ superstructure. Of course it is not possible to implement the $2q$ distortions in this way, but to some degree it should capture the distortions at q . This calculation does reveal new eigenstates at approximately

the right wavevector to account for the backfolded bands observed in ARPES. However, it is not clear to us that this calculation really explains the energetics of the problem – the extra bands we find are basically backfolded versions of the original bands without any clear hybridisation gaps forming – certainly not near EF.

3) As written in the first paragraph, Fermi surface nesting (FSN) is not always the dominant driving force of the CDW. In Ta₂NiSe₇, lattice instability is probably the main source of the modulation with q. While the evaluation of the lattice dynamics is not easy in this complex material, an examination of the importance of FSN is possible by calculating the bare electron susceptibility from the DFT bands. If the susceptibility is presented (even in the supplemental information), the authors can discuss the contributions from the electronic bands more plausibly.

Further calculations are clearly well-motivated: the electronic susceptibility, phonon spectra, electron-phonon coupling, and supercell calculations would all be worthwhile to consider. However, as a team of experimentalists, we use DFT as a tool to help understand the features seen in the ARPES, but feel that we should leave the more advanced calculations to DFT specialists for future work. There are many intricacies of both these more advanced calculations in general, and the material specifically (e.g. low-symmetry space group, C-centered unit cell with 10 inequivalent atoms, small and intricate Fermi surfaces, ...) which combine to make such calculations a significant challenge, and although we were able to take the specific step of doing 1x2x1 supercell calculations in response to the referee's previous comment, we are not in a position to attempt further calculations in new directions. However, our data provides some direct input for such calculations – e.g. showing that using the mBJ potential is an improvement on GGA. Moreover, we would be open to any future collaborations with theorists, outside of the scope of the current work.

4) At first glance, I cannot find the meanings of “BTP”, “OCT”, and “d-OCT” in Fig.1(a). They probably mean “bicapped trigonal prismatic”, “octahedra”, and “distorted octahedra”, respectively, as written in the main text. The authors should put their abbreviations behind them [e.g., octahedra (OCT) in the main text]. We have now explained the acronyms in the main text and the figure caption.

5) The authors should present the value of the temperature used in the ARPES measurements in the figure caption of Fig.2.

We have added this for clarity, although our methods section states “at sample temperatures generally below 10 K except for temperature-dependent runs”

6) On the first line on page 9, the authors mention that both Ta₂NiSe₇ and Ta₂NiSe₅ are semimetals in their high-temperature phases. This is true in the DFT calculations. However, the temperature dependences of the resistivities of Ta₂NiSe₅ and Ta₂NiSe₇ are obviously different, and the gap opening of Ta₂NiSe₅ above T_c is probably under debate [e.g., Nat. Phys. 17, 1024 (2021)]. The author should express this sentence considerably (for example, “both are semimetals”  “both DFT band structures are semimetallic”).

We have adapted the referee's phrasing here and now say “there are similarities in that both are semimetals in their high-temperature phases (at least, according to DFT),...” and we now also cite the mentioned reference. We did not intend to raise a controversial point here.

Reviewer #3 (Remarks to the Author):

Watson et al. present an experimental study of the electronic structure of Ta₂NiSe₇ using high-resolution ARPES which is supported by DFT band calculations. Ta₂NiSe₇ is a less-well-studied CDW material with T_c ≈ 60 K. Its cousin, the “215” compound (Ta₂NiSe₅) however, has been widely studied because it is an excitonic insulator candidate. Here, the authors investigate the CDW in Ta₂NiSe₇ by mapping the Fermi surface and conclude that no states can be directly “nested” onto one another by the expected q-vector, based on basic geometry arguments. Instead, they claim that some portions of FS can be nested by 2q, although they do not observe associated bandgaps opening in the reconstructed phase - a key signature of CDW formation. Curiously, they do however observe a backfolded back with reasonable intensity at q. The overall picture is unusual and slightly confusing, and the authors leave many open questions. Previous ARPES works exist e.g. [Ref 18], although the study by Watson et al. is likely to be the most detailed to date.

Overall, I find the manuscript to be very well-written and clearly presented. The main results however are somewhat speculative, relying on clues in previous works and then interpreting the main features of the electronic structure using geometric arguments related to the CDW i.e. by essentially drawing an arrow (like we see in FIG.4) to test if there are states which are separated by a vector of expected length, q . At a basic level, this is certainly a prerequisite for CDW formation, and we see this type of geometric interpretation in the literature often without deeper analysis. Despite this, the authors do present a detailed and engaging discussion, and have clearly considered several scenarios.

It is a fair statement that our interpretation drawing a link between the electronic structure and the literature result of $2q$ displacements at $2q$ is somewhat speculative. What we would emphasise, though, is that our analysis of the absence of low-energy states nested by q is a robust and important result, that by itself ensures that the CDW in Ta_2NiSe_7 is already rather unusual. For simpler and better-known CDW materials, e.g. TMDCs, it would be fully reasonable to expect deeper analysis than what we have presented, but as the referee notes, this is a complex case that is only now receiving some attention from the electronic structure perspective, and we therefore hope that our early-stage and speculative conclusions are appreciated in this context.

Since the case of Ta_2NiSe_7 seems to be rather complicated (q vs. $2q$), and some effects are hidden to the current data (e.g., gap opening), further calculations of the k -dependent electron-phonon coupling [Phys. Rev. B 77, 165135] or electronic susceptibility [Phys. Rev. B 88, 035108 (2013)] would be informative. In this way, one could analyse potential contributions at both q and $2q$ and learn more about the dominant mechanism.

While we agree with the referee in principle that these would be informative calculations which are clearly called-for, as experimentalists we feel that we should leave these to others for future work, and want to retain an experimentalists perspective in the current paper (see also response to Reviewer #2). Here we use our DFT calculations primarily to understand the origin of the experimental dispersions, rather than seeking to investigate the instability itself from an *ab-initio* perspective. We have added a sentence in the discussion to draw particular attention to the k -dependent electron phonon coupling and electronic susceptibilities, citing the mentioned references.

Related to the hidden gap problem, the authors already touch on the complications of the 3D FS (k -perp.), have they also considered the potential for above- E_f gaps [e.g. Nat. Comms. 12, 6837, (2021)] or is this unlikely in the present situation?

We thank the referee for this interesting suggestion which we have now included in the discussion.

A curiosity: In FIG 3b, at 91.9 K, slightly above T_c , it seems as if there is some weak spectral weight in the region where the backfolded band later appears at 6.2 K. In TiSe_2 (a famous example of intense backfolded CDW bands), spectral weight above T_c was attributed to fluctuations [New J. Phys. 14 075026 (2012)]. On page 8, the authors hint at fluctuations related to q in Ref 16 – are there clues in the authors' data?

For the referee's curiosity, it is true that there is a hint of "precursor" spectral weight at temperatures just above T_c , perhaps analogous to what has been seen in TiSe_2 . However the signal/background is much lower than in TiSe_2 as the backfolding is much weaker, so it is hard to be sure, and we aren't in a position to make a statement about this as we don't have enough systematic data and reproducibility tests.

Final minor comment about the use of colloquial or informal language which are not in-keeping with rest of the paper, listed below:

- Page 1: It is perhaps not appropriate to refer to Peierls' 1955 theory as a "toy model" (despite its obvious simplicity). We have removed this
- Page 5, para 3: "fuzzy": Poorly-resolved? Broadened? Noisy? Poor statistics? These are not only alternative word suggestions, but also have different scientific origins. Perhaps the authors can select the most appropriate option for their supposed k -perp averaging problem.

This was admittedly colloquial language which we have removed. Our understanding is that the main physical effect here is a k_{\perp} broadening, combined with a relatively weak photoemission matrix element for the conduction band which doesn't help the signal/noise.

- Page 6, para 1: "bad underestimation" is subjective. Perhaps "DFT typically underestimates band gaps in semiconductors"

We have adopted the referee's suggestion here.

In summary, the work by Watson et al. represents a comprehensive study of the electronic structure of Ta₂NiSe₇ from both experiments and calculations, with excellent data quality (low temperature, high-resolution, photon energy dependent ARPES). The observation of the backfolded band at q is interesting, and similarly the lack of potential 'nesting' at $2q$ and related bandgap expected for a CDW. Although the main conclusions of the paper are not ground-breaking for understanding the CDW in Ta₂NiSe₇ at this stage, the work is still likely to be important to the community and will motivate further study to answer the questions raised. Based on this, I would be inclined to recommend the work for publication, given that the authors carefully consider my comments.

We highly appreciate the valuable feedback of the referee and are pleased that they are able to recommend it for publication.

REVIEWER COMMENTS

Reviewer #1 (Remarks to the Author):

The manuscript has been modified according to the referee comments and I think it has improved significantly. However, the physical picture of the "backfolding at q but nesting at $2q$ " is still not clear to me. I suggest the authors to show additional experimental evidence such as XRD or STM, or perform DFT calculations for the q -dependent electron susceptibility as the other two referees suggest. Without either of the above two, there is no reason to exclude the final state effect in the photoemission process and the manuscript is not convincing enough for publication in Nat. Comm.

Reviewer #2 (Remarks to the Author):

The authors addressed my suggestions reasonably and some confusing points have been clarified by the revised manuscript. While there are still open questions (e.g., the dominant driving force of the CDW), the experimental results reported in this work can motivate further study in the research community. I would like to recommend this manuscript for publication.

Reviewer #3 (Remarks to the Author):

After consideration of the revised manuscript and response to referees, I believe that the authors give reasonably satisfactory replies to the main concerns.

In the manuscript, the changes are mostly limited to a few sentences for improved clarity or discussion. The authors present some additional results, such as the $1 \times 2 \times 1$ supercell calculations and XRD data in the supplementary, but these do not significantly strengthen the claims of the paper.

A concern shared among referees was that there is not sufficient evidence (or understanding of) the peculiar q vs $2q$ scenario at this stage. I would also add the lack observation of the CDW bandgap. The referees collectively called for further experiments and/or more detailed DFT calculations focusing on the CDW instability (e.g., electronic susceptibility). Further data or analysis was not

provided by the authors, although they made efforts to justify their reasons for this decision in the response letter. They openly admit that they are not able to fully explain their results, and rather focus on the novelty of their experimental observation (presented as-is) to motivate further study. They say that they do not have the expertise to perform more complex DFT calculations, which is fair.

Overall, the main experimental (ARPES) results are of high quality and the analysis is sound. Although the authors were not able to fully explain the peculiar CDW scenario, I believe that the simple presentation of the unexpected discovery is okay in this case and will certainly generate curiosity in the community. The authors have clearly considered the questions surrounding their work given the detailed discussion section.

I stand by my opinion of the first round of review, and I recommend the work for publication.